# Placental Volume, Vascularization, and Epigenetic Modifications in Obesity and Gestational Diabetes: A 3-D Ultrasound and Molecular Analysis

**DOI:** 10.3390/life15111691

**Published:** 2025-10-30

**Authors:** Balint Kolcsar, Kata Kira Kemeny, Zoltan Kozinszky, Eszter Ducza, Andrea Suranyi

**Affiliations:** 1Department of Obstetrics and Gynecology, Albert Szent-Györgyi Medical School, University of Szeged, H-6725 Szeged, Hungary; kolcsar.balint@med.u-szeged.hu (B.K.); kozinszky@gmail.com (Z.K.); 2Department of Pharmacodynamics and Biopharmacy, Faculty of Pharmacy, University of Szeged, H-6720 Szeged, Hungary; kemeny.kata.kira@szte.hu (K.K.K.); ducza.eszter@szte.hu (E.D.); 3Capio Specialized Center for Gynecology, Solna, 171 45, Stockholm, Sweden

**Keywords:** apelin, leptin, VEGF, DNA methylation, pregnancy, obesity, placenta, gestational diabetes

## Abstract

Background: Obesity and gestational diabetes mellitus (GDM) are the most common metabolic conditions that have an unfavorable impact on maternal and fetal health. Maternal obesity and GDM are often associated with placental dysfunction and structural alterations. The apelin receptor (APLNR), vascular endothelial growth factor (VEGF), leptin, and DNA methylation play crucial roles in placental function. We aimed to investigate the placental volume and vascularization, and to determine the changes in these markers in obese and GDM mothers. Material and Methods: In our study, we investigated the human placenta (*n* = 48) at term. The placental structural analyses on volume and vascularization were conducted using three-dimensional ultrasound before labor. Placental APLNR expression was determined using RT-PCR, and leptin and VEGF concentrations using ELISA in placental tissues. Global DNA methylation was measured using colometric assay. Results: The age of GDM mothers was significantly higher than that of normal and obese mothers. The gestation length of GDM mothers was significantly shorter than that of normal and obese mothers. The placental volume was significantly higher in obese and GDM cases compared with normal cases. Vascularization indices (VI, FI, VFI) were significantly depressed in GDM and obesity. In the case of biomarker studies, APLNR, leptin, and VEGF showed similar decreases in obese and GDM placentas. Based on our results, the effect of GDM, not obesity, was more pronounced for these biomarkers. VEGF reduction correlates with three-dimensional placental vascularity studies. The DNA methylation was significantly elevated in both GDM and obese placental samples, while the GDM effect was more pronounced. Conclusions: This study is the first to demonstrate structural alterations of the placenta using placental tissue biomarkers in obesity and gestational diabetes mellitus (GDM). We found that both GDM and obesity affect placental volume and vascularity, as indicated by reduced leptin and VEGF levels, presumably mediated by epigenetic effects. Our findings may provide a novel therapeutic target for improving abnormal placental function caused by GDM and obesity.

## 1. Introduction

Gestational diabetes mellitus (GDM) is a certain type of diabetes that is diagnosed during pregnancy. GDM is defined as high blood glucose levels during pregnancy in women who have never previously experienced insulin sensitivity issues. This condition can develop because pregnancy hormones make the cells less efficient at using insulin, the hormone that regulates blood sugar levels [1]. GDM occurs in about 10% of all pregnancies in Europe [1], making it the most common medical complication during gestation [2,3,4]. In recent decades, the incidence of GDM has increased worldwide. The pooled international standardized prevalence of GDM was 14.0%. The regional standardized prevalence of GDM was 7.1% in North America and the Caribbean area, 7.8% in Europe, 10.4% in South and Central America, 14.2% in Africa, 14.7% in the Western Pacific, 20.8% in Southeast Asia, and 27.6% in the Middle East and North Africa. The standardized prevalence of GDM in low-, middle-, and high-income countries was 12.7%, 9.2%, and 14.2%, respectively [5].

GDM and obesity are interrelated conditions that pose a significant health risk to both mothers and newborn babies. The incidence of maternal obesity is exponentially rising all over the world and is an important obstetric risk factor. Increasing rates of obesity among women of childbearing age are contributing to the growing prevalence of GDM, reflecting patterns observed across Europe. Factors such as urbanization, sedentary lifestyles, and dietary changes are pivotal drivers. This escalation poses significant public health challenges, as both obesity and GDM are associated with adverse maternal and infant outcomes, including an increased risk of metabolic disorders, cardiovascular disease, and complications during pregnancy and delivery. The WHO’s annual reports show that 43% of women were overweight and 16% obese in 2022 [6]. GDM and obesity are influenced by several biochemical and molecular factors.

Apelin (APLN) is a newly identified adipokine, whose receptor, apelin receptor (AP), is a member of the GPCR family [7,8]. AP does not bind angiotensin II despite having a transmembrane domain that is 54% similar to that of the angiotensin type 1 receptor (AT1R) [8]. Thus, it was long regarded as an orphan receptor until Tatemoto et al. [9] identified apelin, the first endogenous ligand for AP, which signaled the change from an orphan to a multifunctional receptor [8]. Apelin and apelin receptors are expressed in many tissues and organs, including reproductive tissues. The important role of apelin in various pathological conditions (e.g., diabetes, obesity, cardiovascular disease, and PCOS) has been demonstrated [7]. This peptide is involved in glucose regulation, cardiovascular function, and angiogenesis. It has elevated levels in obesity and gestational diabetes, and enhances glucose uptake. Apelin may improve insulin sensitivity, potentially aiding glucose tolerance [10].

Vascular endothelial growth factor (VEGF) is a potent factor in angiogenesis, playing an essential role in the formation of new blood vessels, which is vital for tissue growth and repair [11,12]. It was first discovered in vascular endothelial cells as a growth factor [11]. It is crucial in the context of placental development [12]. In obesity and gestational diabetes (GDM), elevated VEGF can cause abnormal angiogenesis, leading to complications like fetal growth restriction [13]. Hypoxia in adipose tissue triggers the overexpression of VEGF, exacerbating inflammation, insulin resistance, and metabolic complications [12,13].

Leptin is a peptide hormone that controls body weight, dietary intake, and reproductive processes. After being synthesized and secreted by adipose cells of white adipose tissue, leptin, which is a result of the obese (ob) gene, attaches to and activates its receptor, the leptin receptor (LEP-R) [14]. In obesity and GDM, elevated leptin levels lead to leptin resistance, reducing its effectiveness. This resistance may disrupt energy and glucose balance, affect placental function, promote fetal overgrowth, and contribute to chronic inflammation, worsening insulin sensitivity and metabolic dysregulation [14,15].

DNA methylation, an epigenetic modification that adds methyl groups to DNA, typically represses gene expression [16]. Influenced by factors such as diet and metabolism, abnormal methylation is associated with obesity and gestational diabetes mellitus (GDM), affecting genes involved in glucose and lipid metabolism, inflammation, and insulin signaling [17,18]. These changes can influence offspring’s health, increasing the risk of future metabolic disorders [16,19].

GDM and obesity share overlapping pathophysiological mechanisms, including insulin resistance, chronic inflammation, and hormonal dysregulation. In GDM, placental hormones exacerbate pre-existing insulin resistance, impairing glucose uptake. Obesity contributes to excess adiposity, which reduces adiponectin and increases pro-inflammatory factors, thereby disrupting insulin signaling [16,17,18,19]. These mechanisms highlight targets for therapies, including weight management, anti-inflammatory agents, incretin-based drugs, and interventions to restore adipokine balance. Therefore, we selected vasculogenic factors, including apelin receptor (APLNR) expression and VEGF, as well as glucose regulator markers, APLNR and leptin, and DNA methylation, which is influenced by diet and glucose metabolism. The interaction between APLNR, VEGF, leptin, and DNA methylation creates a complex network in the placenta that influences the development and progression of GDM and obesity. An inflammatory state driven by leptin and VEGF can alter DNA methylation patterns, which in turn can affect the expression of genes involved in metabolism and placental vascular function. Apelin’s role in modulating insulin sensitivity and glucose uptake is further integrated into this placental network, potentially offering a protective effect against the metabolic abnormalities observed in these conditions [14]. It is essential to understand these interactions to develop novel therapeutic strategies. For example, targeting leptin resistance or VEGF pathways could potentially improve some of the complications associated with GDM and obesity. Similarly, interventions aimed at modifying epigenetic traits, for example, through diet or pharmacological agents, hold promise in mitigating the long-term effects of these conditions.

The aim of our study was to determine whether placental 3-D ultrasound and placental biomarker analyses are related to obesity and GDM, and whether the differences between the two pathologies are similar near labor. Therefore, we analyzed the complex network interconnection of placental biomarkers, the placental morphologic features, and the clinical outcome of GDM and obesity close to term.

## 2. Materials and Methods

### 2.1. Samples

This was a prospective, cross-sectional cohort study carried out at the Obstetrics and Gynecology Clinic, the University of Szeged, Hungary, recruiting pregnant women who delivered by cesarean section between June 2023 and March 2024. The registered gestational age of the subjects ranged between the 34th and 41st week (*n* = 48). Placenta samples (cc. 1 cm^3^) were gathered during the operative deliveries of the participants. The study population contained healthy controls (*n* = 30), GDM (*n* = 6), and obese pregnant women (*n* = 12 with a BMI > 30 kg/m^2^). GDM was diagnosed using WHO guidelines. The study protocol was approved by the Clinical Research Ethics Committee of the University of Szeged (reference number: 57/2020-SZTE). The study was conducted in accordance with the principles of the Declaration of Helsinki. Written informed consent was obtained from all participants.

The inclusion criteria for this study were as follows: single pregnancy without any fetal malformation and normally located placenta. BMI calculation was completed in early pregnancy for obesity diagnosis (BMI > 30 kg/m^2^). According to the recommendation of the World Health Organization, GDM is a pathological condition in which women without previously diagnosed diabetes exhibit high blood glucose levels during pregnancy, which disappears or maintains after pregnancy. The diagnostic criteria for GDM are based on the World Health Organization’s guide: screening by 100 g oral glucose test between 24 and 28 weeks of gestation.

The exclusion criteria for this study were as follows: overweight women (BMI = 25–30 kg/m^2^), undernourished women (BMI < 18 kg/m^2^), multiple pregnancy, structural or genetic abnormalities of the fetus and/or newborn, pathologic localization of placenta (e.g., placenta praevia), abnormal placentation (placenta accreta spectrum), and self-reported drug, alcohol, or nicotine addiction. GDM accompanied by another systemic disease (autoimmune disease, vasculitis, hemophilia, thrombophilia, HIV infection, etc.) was also an exclusion criterion, as well as obesity accompanied by another disease (GDM, hypertension, pre-eclamisia, etc.).

Gestational age was established from the menstrual history and confirmed from the measurement of fetal coronal apex length at the first trimester scan.

Placental tissue sampling for cryopreservation at delivery involves the collection of small sections of the placenta immediately after childbirth. This process was used for research only. Right after the newborn is delivered and the placenta is expelled, the head of the medical team prepares sterile tools for tissue collection. The placenta was examined to ensure it was intact and suitable for sampling. Specific areas of the placenta are chosen for sampling, targeting the villous tissue. Tissue samples can be taken from different regions to assess varying structures or conditions. A sterile scalpel was used to excise small pieces (cc. 1–2 cm^3^) from the placenta. Care is taken to avoid contamination. The tissue samples are placed in sterile tubes. The tubes were clearly labeled with identifying information, such as the mother’s name, date, and specific tissue location. We put tubes in secondary containment, biohazard plastic bags. During external shipping, a rigid, tertiary insulated box ensured safety during transport at temperatures between 2 °C and 8 °C. The transportation time was less than 10 min. The proper transportation of samples ensured compliance with biosafety standards. The tissues were collected in RNAlater Solution (Sigma-Aldrich, Hungary), frozen in liquid nitrogen, and stored at −80 °C until the extraction.

### 2.2. Ultrasound Investigations

All ultrasound examinations were performed transabdominally 16–48 h before the cesarean section. Standard fetal and placental parameters were assessed by a routine two-dimensional (2-D) scan. The factorial standard setting ‘Obstetrics/2–3 trimester’ was applied for 2-D sonographic scans. Afterwards, a three-dimensional (3-D) sweep was acquired across the placenta with power Doppler settings. The 3-DPD sweep was achieved through the placenta with a Voluson S10 BT2021 ultrasound machine (RAB 2–5 MHz probe and 4D View version 10.4 program; GE Healthcare, Kretztechnik, Zipf, Austria). The placental volume was obtained at maximum quality with a time elapsed between 10 and 15 s, using a perpendicular insonation angle to the placental plate. The same ultrasound preset was used for all subjects (power, 96%; frequency, low; quality, normal, density, 6, ensemble, 16; balance, 150; filter, 2; smooth, 3/5; pulse repetition frequency, 0.9 kHz, and gain, −0.2). Each 3-D image was retrieved from the disk for further evaluation procedures. The sonographic pictures were processed employing the virtual organ computer-aided analysis (VOCAL) program connected to the computer software 4D VIEW (GE Medical Systems, Austria, version 10.4), which consists of drawing the outline of the placenta repetitively after rotating its image 6 times by 30°, thus preventing the inclusion of decidua and maternal vessels. After the completion of the 180°rotation, the placental volume was automatically calculated by the VOCAL software (GE Medical Systems, Austria, version 10.4). For each patient, placental volumes were measured 3 times by a trained and experienced ultrasound specialist. The 3-D volume pictures are compiled of voxels as small units of volume, including all data concerning gray and color intensity scales ranging from 0 to 100. Based on these values, 3-D power Doppler indices can be received by analyzing voxels processed by the ultrasound software system (Version 10.4), which describes the vessel density and blood flow in the placenta. These 3-DPD indices are utilized in evaluating placental perfusion, and it is broadly recognized that they might display both uteroplacental and fetoplacental blood flow.

The vascularization index (VI), defined as the ratio of color voxels to total voxels, quantifies the presence of blood vessels within a given volume of interest and is expressed as a percentage (vascularity). The flow index (FI) represents the average color value of all color voxels, indicating the average intensity of blood flow on a scale from 0 to 100 (unitless). The vascularization flow index (VFI), calculated as the weighted ratio of color voxels to total voxels, merges information on vessel presence (vascularity) and the quantity of blood cells transported (unitless). The values range from 0 to 100. The VOCAL program calculates the 3-D power Doppler (3-DPD) indices (VI, FI, VFI) on the ground of the gathered samples.

### 2.3. RT-PCR Studies

#### 2.3.1. Total RNA Preparation from Tissue

The placental samples (1–2 cm^3^) were mechanically ground by Sartorius MikroDismembrator U (Sartorius, Göttingen, Germany). The grinding was performed at room temperature. The total cellular RNA was extracted from ground placenta tissues by guanidinium thiocyanate–acid–phenol–chloroform according to the procedure of Chomczynski and Sacchi [20]. Following precipitation with isopropanol, the RNA was washed with 75% ethanol and then resuspended in diethylpyrocarbonate-treated water. The purity of the RNA was verified by measuring the optical density at 260/280 nm using BioSpec Nano (Shimadzu, Kyoto, Japan), with all samples showing an absorbance ratio within the range of 1.6–2.0. The quality and integrity of the RNA were evaluated through agarose gel electrophoresis.

#### 2.3.2. Real-Time Reverse Transcription Polymerase Chain Reaction (RT-PCR)

The reverse transcription polymerase chain reactions were conducted utilizing the TaqMan RNA-to-CT-Step One Kit (Thermo Fisher Scientific, Budapest, Hungary) in conjunction with an ABI StepOne Real-Time cycler. Placenta tissues were collected and subsequently immersed in RNAlater Solution (Sigma-Aldrich, Budapest, Hungary). The samples were stored at −80 °C until total RNA extraction was performed. The preparation of total RNA and the real-time polymerase chain reactions were conducted following our previously published experiments [21]. The primers used included the Hs00270873_s1 assay for the apelin receptor (APLNR) and the Hs01060665_g1 assay for β-actin as an endogenous control (Thermo Fisher Scientific, Budapest, Hungary).

### 2.4. Enzyme-Linked Immunosorbent Assays (ELISAs)

The placental tissues were powdered and homogenized with a solution of RIPA lysis and extraction buffer and a protease inhibitor cocktail. Following protein extraction, the protein concentrations in the supernatant layer were measured with a spectrophotometer. Leptin and VEGF concentrations were determined using an ELISA assay kit (Human Lep ELISA Kit, Human VEGF kit, FineTest, Wuhan, China). ELISA plates were read by a SPECTROStar Nano spectrophotometer (BMG Labtech, Ortenberg, Germany).

### 2.5. Global DNA Methylation

DNA isolation from placental tissues was performed using the GeneaidTM DNA Isolation Kit (Geneaid Biotech Ltd., New Taipei City, Taiwan), and global DNA methylation was assessed using the Methylated DNA Quantification Kit (Abnova Ltd., New Taipei City, Taiwan).

### 2.6. Statistical Analysis

Statistical analyses were assessed utilizing Prism 10.2.1 software (GraphPad Software Inc., San Diego, CA, USA, 2024). All statistical calculations were evaluated with a one-way analysis of variance (ANOVA) (corrected with Dunnett’s post hoc test) or unpaired t-test where appropriate, and each is presented as the mean ±statistical error of mean (SEM). A significance level of *p* < 0.05 was accepted.

## 3. Results

### 3.1. Changes in Placental Volume and Vascularization

In our study, maternal age was significantly higher in the group of GDM pregnant women compared with the control and obese mothers (Table 1). The GDM cases had a BMI within the normal range (18–25 kg/m^2^).

Table 2 exhibits the values of the placental perfusion using the three-dimensional power Doppler indices. The placental volumetry was prepared in 3-D mode by the VOCAL program (Figure 1). The in vivo vascular measurements were performed immediately below the umbilical insertion, which is the most densely vascularized part of the placenta (Figure 2). The obese and GDM participants expressed significant differences in terms of the sonographic placental parameters. The placental volume was significantly elevated, while the vascularization indices (VI: vascularization index, FI: flow index, and VFI: vascularization flow index) were significantly depressed compared with normal-weighted pregnant women. There were no significant alterations in placental volume and 3-D PD indices between the obese and GDM cases.

### 3.2. Changes in Apelin Receptor mRNA Expression

Apelin receptor mRNA expression was significantly diminished in the samples from women with GDM; however, the decrease was not significant in the samples from obese women compared with the control samples (Figure 3).

### 3.3. Changes in VEGF and Leptin Placental Concentration

The VEGF concentration decreased significantly in the GDM and obese samples compared with the control in the placental tissue (Figure 4).

Leptin concentration was significantly decreased in placental samples from GDM and obese patients, but this decrease was significant only in the GDM group (Figure 5).

### 3.4. Changes in Placental DNA Methylation

The 5-mC (5-methylcytosine) percentage of the GDM and obese samples increased significantly compared with the control (Figure 6).

## 4. Discussion

Although the prevalence of diabetes is increasing rapidly worldwide [4], the prevalence estimation and its regional variation in the case of various types of diabetes, particularly GDM, is challenging due to wide disparities in screening modalities and diagnostic principles applied to recognize the disease. Fetal abnormalities, such as heart failure [4], are at least 15 times more common than in the children of non-diabetic pregnancies [4]. In addition to the short-term neonatal, fetal, and maternal concerns related to GDM, there are also long-term sequelae for the woman and her offspring [4]. While research on adult offspring is restricted and inconclusive due to great variations in study populations, there are many human studies in children from different populations with different types of maternal diabetes [22]. The incidence of GDM has increased over the past decade as the population’s BMI has increased by 12% among women of reproductive age [23]. In our study, we examined these two high-risk groups.

In our ultrasound-based studies of singleton pregnancies, placental volume and weight were higher in obese and GDM pregnancies than in normal-weight controls. Most studies present data only after birth. Our study is the first to present in utero data on placental volume and vascularization close to term, which is consistent with postnatal confirmed placentomegaly. Previous studies have reported that obese and GDM mothers have an augmented predisposition to develop placentomegaly due to impaired vasculogenesis and angiogenesis, leading to edema [24,25]. Our data confirmed that third trimester placentomegaly and placental volume were not significantly different between the GDM and the obese cases.

We observed a negative correlation between maternal BMI and 3-D PD indices in pregnancies complicated by obesity and GDM. Compared with the control group, the indices were reduced in the pathological groups. Although the VI, FI, and VFI values overlapped in the GDM and obese groups, any larger placental volumes were associated with lower vascular indices. The increased placental volume expansion is related not only to BMI, but also to GDM. In obesity and GDM, decreased placental VI and VFI were linked with inadequate angiogenesis and unfavorably declined arteriolar number per volume, causing placental edema. However, a depressed FI may mean a narrower internal vessel diameter. The 3-D ultrasound examination of the placental perfusion did not show any difference between the obese and GDM groups, while the biomarkers showed characteristic shifts [26]. It has been proposed that the placenta has a critical role in mediating inflammation processes in obese women with GDM. Placental structure and function may change during the adaptive response to obesity, and the placenta may serve as a target and source for inflammatory cytokines and biomarkers during pregnancy [27].

During pregnancy, hormonal changes certainly increase insulin resistance, which ensures that more glucose is available for the fetus [28]. In women who already have insulin resistance prior to pregnancy (often associated with obesity), this additional insulin resistance can overwhelm the pancreas’s ability to produce insulin, leading to elevated blood sugar and GDM [28,29]. Inflammation is also manifested by an increased metabolic rate due to a concentrated and rapid response to an insult [27]. A profile of immune cells favors a pro-inflammatory environment in tissues such as adipose, the pancreas, and the liver; it is chronically maintained by metabolic cells, such as adipocytes, without resolution and is associated with a reduced metabolic rate [29,30,31]. For these reasons, the term “metaflammation” was coined to describe the special immunophenotype associated with obesity [30]. The link between obesity, inflammation, and insulin resistance was first discovered when it was observed that the adipose tissue of obese individuals emitted increased levels of pro-inflammatory cytokines, and that antagonism led to increased insulin sensitivity [29].

Weight gain might be associated with an increase in cytokines and has an effect on insulin resistance formation. It is clearly evident that maternal organs become increasingly insensitive to insulin, marked by a 50% reduction in insulin-mediated glucose clearance, and a ∼250% increase in insulin production to prevent hyperglycemic excursions [28,29].

During gestation, molecular processes that support placentation replace the non-pregnancy-related inflammatory profile, as a tightly controlled balance between pro- and anti-inflammatory biosubstrates for implantation, trophoblast invasion, and migration [32]. Pathological placentation caused increased APLNR expression and leptin concentrations in our placental samples from GDM women compared with normal placentas. Obesity has a similar effect, but the decrease in leptin and APLNR expression was not significant.

Our study confirms that GDM is characterized by insulin resistance. Insulin resistance can impair signaling pathways that regulate APLNR expression and leptin secretion. GDM is associated with increased inflammation and oxidative stress, which can lead to changes in the expression of genes and proteins involved in metabolic regulation. These conditions were detected by reducing APLNR expression and leptin production. In GDM, placental function is often devastated, which can affect the production and regulation of hormones and proteins. It was supported by alterations in placental APLNR and leptin.

Together, these factors may lead to the observed decrease in APLNR expression and leptin concentration in women with GDM compared with women without the disease [24,26]. Changes in adipokine release can lead to changes in glucose homeostasis during gestation, and a disequilibrium in adipokines. This is also connected to the pathogenesis of GDM and obesity, and thus, adipokines have attracted significant research interest in these diseases [29]. The concentration of leptin in peripheral blood increased during obesity and GDM, while adipokine concentration decreased in obese [33] and GDM samples [29,33]. A significantly diminished leptin concentration has been observed in obese samples in early pregnancy [33]. However, the leptin concentration increases gradually throughout pregnancy [33]. We detected a decline in concentration in both obese and GDM placental tissues compared with normal placental samples at term. Therefore, we concluded that the decrease in placental leptin concentration in obese patients, which is even more pronounced in GDM patients, is the compensatory mechanism of the placenta to maintain the serum leptin level.

Leptin enhances the synthesis of vascular endothelial growth factor (VEGF). In vitro studies demonstrated the critical role of the interaction between endothelial cells and adipocytes, with hypoxia serving as a factor that amplifies leptin’s effects on VEGF. Genetic and epigenetic mechanisms, along with procreator–offspring programming, may play a role in leptin-VEGF crosstalk. Some female-specific characteristics of the leptin-VEGF relationship in obesity have been observed [34]. In our study, VEGF results, combined with 3-D power Doppler analyses, indicated that vasculogenesis in the placenta was impaired. This placental depression can contribute to placental dysfunction and vascular complications, exacerbating both conditions. Obesity has the same detrimental effect on placental vasculogenesis as GDM. The leptin concentration showed a strong correlation with the VEGF value. Moreover, in GDM, VEGF level depression was 50 percent, and leptin concentration was reduced by 64 percent compared with the normal placental tissue value. Furthermore, in obesity, VEGF level depression was 53 percent, and leptin concentration was reduced by 54 percent compared with the normal placental tissue value. A reduced concentration of VEGF in the placenta can have a significant impact on pregnancy and fetal development. VEGF is a critical regulator of angiogenesis, the process by which new blood vessels form, and is essential for ensuring an adequate blood supply to the developing fetus [33].

The reduced VEGF levels in the placenta are associated with impaired angiogenesis, placental dysfunction, and hypoxia. Possible long-term effects of insufficient placental VEGF include a decline in placental vascularization and edema, resulting in poor placental function and placentomegaly. Our results showed that VEGF decline was significantly associated with both obesity and GDM. In any case, obesity led to a greater inclination in VEGF concentration, and GDM led to a greater inclination in placental function and edema. A decreased VEGF concentration in the placenta may be a critical marker indicating underlying problems with placental function. This necessitates careful monitoring.

DNA methylation, as one of the most prevalent epigenetic modifications, has attracted considerable attention in recent years [33]. In our study, DNA methylation was significantly higher in GDM and obese patients compared with the control. This can be explained by the specific diabetes-related metabolic and hormonal modifications during pregnancy. These changes may affect gene expression differently than obesity alone. GDM-specific factors, such as altered insulin and glucose levels, hormonal fluctuations, and inflammation, can lead to various epigenetic modifications. These modifications may affect the genes involved in glucose metabolism, subacute inflammation, and placental function, resulting in more pronounced DNA methylation changes in GDM than in obesity, which primarily represents chronic metabolic stress.

Population-based screening programs would be ideal to prevent the progression of GDM to diabetes, but unfortunately, epidemiologic screening programs are predominantly absent in routine clinical practice worldwide. This study presents potential alternatives for early screening and follow-up in GDM and obesity. If not treated properly, GDM and obesity can lead to short- and long-term complications for both the mother and the offspring. It increases the risk of maternal hypertension and preeclampsia during gestation. For the neonate, it can lead to an excessive birth weight (macrosomia), premature birth, respiratory distress syndrome, and an increased risk of fetal malformations (e.g., fetal heart defects) and obesity and/or type 2 diabetes later in life.

A limitation of our study was the relatively small sample size. The recruitment of the study participants was partly difficult because obesity is a common underlying condition in GDM. It was considered important not to study obesity cases with concomitant GDM. This allowed us to isolate the effects of GDM and obesity as best as possible. Although associated with a normal BMI, the number of GDM cases is still much lower than the number of obesity cases.

## 5. Conclusions

GDM and obesity are multifactorial conditions influenced by various biochemical and molecular factors, such as apelin, VEGF, leptin, and DNA methylation. This study is the first to conduct a comprehensive clinical trial assessing these biomarkers simultaneously in placental tissue and through sonographic features of the placenta.

A notable strength of this research lies in its clear separation of obese and gestational diabetes mellitus (GDM) groups, allowing for a precise and independent analysis of each condition. The inclusion of appropriate control groups further strengthens this study by enabling the accurate assessment of placental biomarker concentrations at near-term gestational age. These biomarkers are not only essential for elucidating the underlying pathophysiological mechanisms of GDM and obesity but also represent promising candidates for future therapeutic interventions. Importantly, this study emphasizes that measures of adiposity—such as gestational weight gain—should be considered when predicting GDM risk, rather than relying solely on body mass index (BMI) at the initial prenatal visit. This approach underscores the multifactorial nature of metabolic complications during pregnancy. Continued investigation into the complex interactions and regulatory roles of these molecular markers will be vital for advancing our understanding of metabolic adaptations in pregnancy. Such research holds the potential to inform the development of more effective prevention and treatment strategies for GDM in the context of maternal obesity, ultimately contributing to improved health outcomes for both mothers and their offspring.

Obesity and GDM result in a significantly larger placental volume and worse placental vascularization, which can be effectively monitored by changes in VEGF concentration. Insulin resistance may impair signaling pathways that regulate apelin receptor expression and leptin secretion. These factors can lead to the observed decrease in apelin receptor mRNA expression and leptin concentration, which may predict the outcome of insulin resistance and poor glucose metabolism before the manifestation of GDM.

## 6. Limitation

A limitation of this study is that 3-D ultrasound uses expensive software and is not very common in developing countries. Consent for placental sampling at the time of delivery significantly reduced the number of participants because many women did not agree to the sampling.

## Figures and Tables

**Figure 1 life-15-01691-f001:**
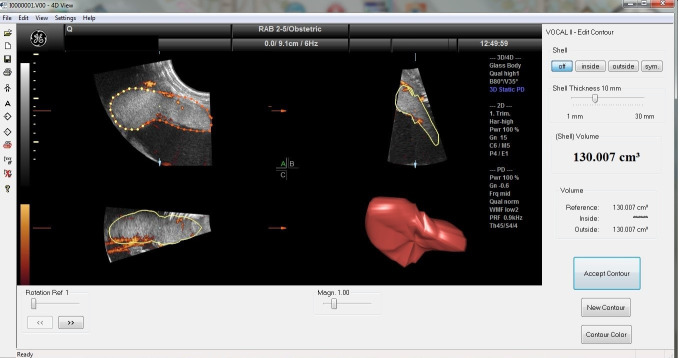
Placenta volumetry in 3-D mode. The multiplanar technique was applied for volume analyses. The dotted area (plan ‘A’) is the border of placental tissue. The yellow line on the plane ‘B’ and ‘C’ represent the border of the placenta. The 3-dimensional figure of the placenta is in the bottom-right quadrant.

**Figure 2 life-15-01691-f002:**
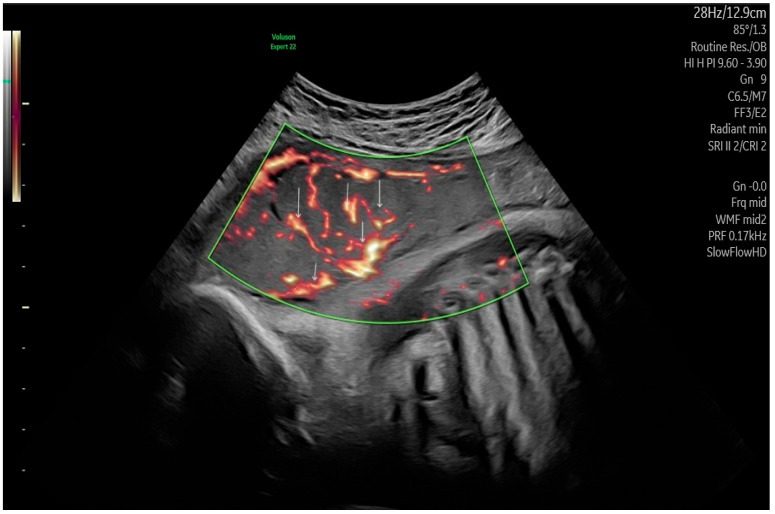
Placental vessel using three-dimensional power Doppler (3-DPD) mode, which facilitates the visualization of vessels (red lines) with a low diameter (↓) with a relatively low-grade velocity blood *flow (in the green box)*.

**Figure 3 life-15-01691-f003:**
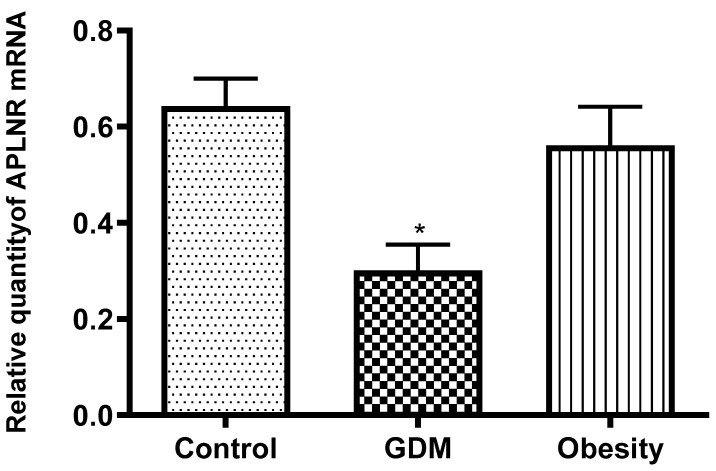
Changes in apelin receptor (APLNR) mRNA expression in the placental tissue samples collected from obese, GDM (gestational diabetes mellitus), and control women; *: *p* < 0.05 compared to the control.

**Figure 4 life-15-01691-f004:**
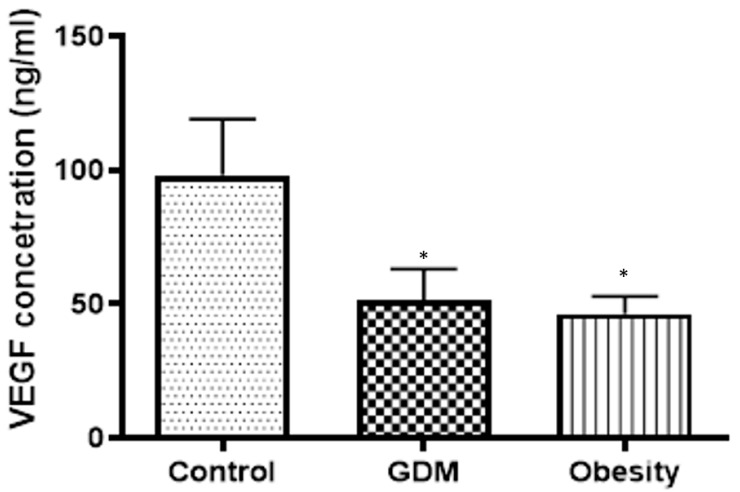
Changes in the VEGF concentration in the placental tissues collected from obese, GDM (gestational diabetes mellitus), and control women; *: *p* < 0.05 compared to the control.

**Figure 5 life-15-01691-f005:**
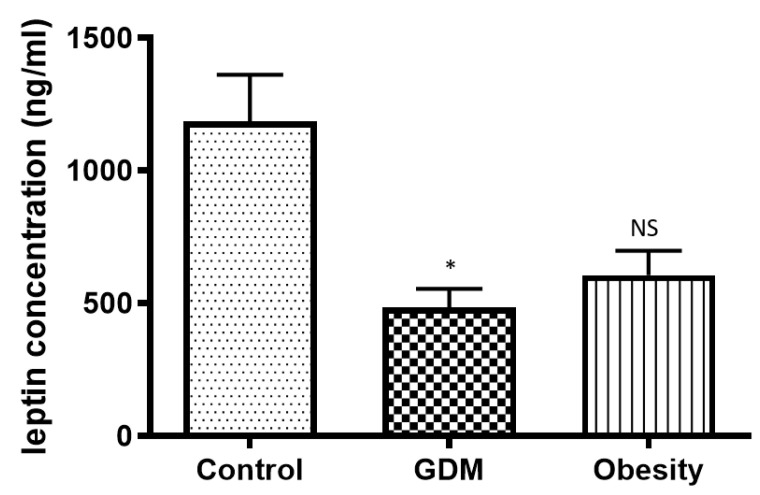
Changes in the leptin concentration in the placental tissue samples collected from obese, GDM, and control women; NS: *p* > 0.05; *: *p* < 0.05 compared to the control.

**Figure 6 life-15-01691-f006:**
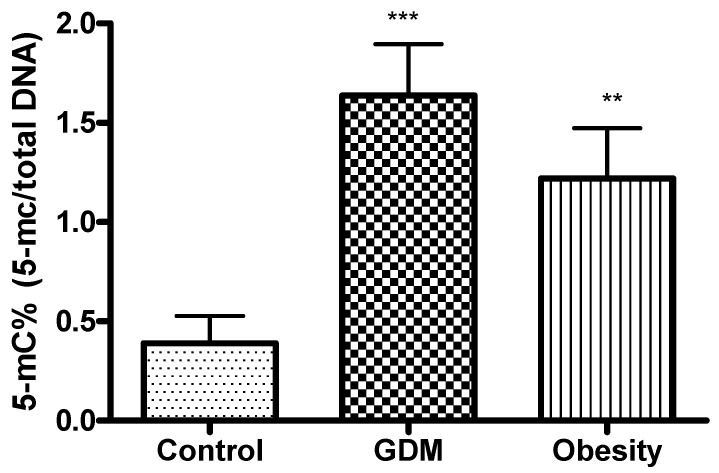
Results of DNA methylation assays in the placental tissue samples collected from obese, GDM, and control women; **: *p* < 0.01; *** *p* < 0.001 compared to the control.

**Table 1 life-15-01691-t001:** Maternal age data of the study groups. ns: *p* > 0.05, *: *p* < 0.05; compared to pregnant women with normal BMI.

Pregnancy	Maternal Age [mean ± SD]	Gestation Age [mean ± SD]
Normal BMI (*n* = 30)	26.6 ± 4	40.5 ± 0.84
BMI > 30 kg/m^2^ (*n* = 12)	29.5 ± 2.4 ^ns^	39.5 ± 2.1 ^ns^
GDM (*n* = 6)	35.6 ± 3.2 *	36.9 ± 1.36 *

**Table 2 life-15-01691-t002:** Results of placental sonography.VI: vascularization index, FI: flow index, and VFI: vascularization flow index. *: *p* < 0.05, compared to pregnant women with normal body mass index (BMI).

Placental Sonography	Normal BMI (*n* = 30)	BMI > 30 kg/m^2^ (*n* = 12)	GDM (*n* = 6)
Placental volume (ml ± SD)	527.3 ± 93.1	775.6 ± 143.2 *	754.6 ± 155.3 *
VI (mean ± SD)	14.11 ± 5.1	8.71 ± 2.4 *	7.67 ± 3.3 *
FI (mean ± SD)	44.97 ± 22.64	37.4 ± 10.9 *	39.4 ± 14.1 *
VFI (mean ± SD)	8.21 ± 3.63	4.74 ± 1.34 *	3.99 ± 2.67 *

## Data Availability

The original contributions presented in this study are included in the article. Further inquiries can be directed to the corresponding author.

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
