# Peer review of "Placental Volume, Vascularization, and Epigenetic Modifications in Obesity and Gestational Diabetes: A 3-D Ultrasound and Molecular Analysis"

_life, 2025, doi:10.3390/life15111691_

Round 1

Reviewer 1 Report

Comments and Suggestions for Authors

The Authors have well designed this study and executed well but number of samples is very low to justify the finding.

  1. The Sample number is very low, GDM=6, Obesity = 12, 30 was control, these low number of sampling may mislead the study findings.
  2. If it is a diagnostic biomarker approach, the study should be designed to diagnose in the early trimester and before delivery.
  3. 3D ultrasound is very expensive and it is not very common in developing countries.
  4. Inclusion and exclusion criteria (IC&EC) could have been well defined IC & EC may be presented as separate paragraph
  5. The size of the placenta sample is so small, is it enough to do proposed study?
  6. Have you got signature in the informed consent forms?
  7. Any questionnaire or study information shared among the pregnant women?
  8. In the line no 219 authors says “The placental tissues were powdered”, why it was dried? Sun dried? Any chance of loss of protein by over heating?
  9. In the figure 3., Changes in the apelin receptor (APLNR) mRNA expression is not very significantly high or low
  10. The title, objective & conclusion can have same study content but there are few mismatch

Author Response

Dear Reviewer 1,

We have found the comments of the Reviewer 1 very useful, and we hope that we can manage to reply to them satisfactorily and the changes we have made will meet with the approval of the reviewers.

The point-by-point responses are in attached file.

Reviewer 2 Report

Comments and Suggestions for Authors

‘Placental Volume, Vascularization, and Epigenetic Modifications in Obesity and Gestational Diabetes: A 3D Ultrasound and Molecular Analysis’ is original research article conducted with the aim to investigate the placental volume and vascularization, and to determine the changes in markers (leptin, VEGF, APLNR, DNA methylation) in the obese and GDM mothers.

The manuscript is clear, relevant to the field and presented in a well-structured manner. Majority of cited references are mostly recent publications and relevant. The ethics statements and data availability statements are adequate. The manuscript’s results are reproducible based on the details given in the methods section.

This study is the first to conduct a comprehensive clinical trial testing mentioned biomarkers simultaneously in placental tissue and through sonographic features of the placenta and this is the biggest value of this study.

I have concerns regarding the design of the study. Inclusion criteria for cases and controls are not clearly stated. Please add this to your revised manuscript. The number of analyzed samples with GDM (6) and obesity (12) is rather small. This should be stated as a limitation of the study. Since the number of tested samples is small, it is questionable whether conclusions are supported by the results. In my opinion, since GDM and obesity among pregnant women are not that rare, authors should have collected and analyzed more placenta samples. Or provide strong explanation why it is not done in a first place.

I recommend this manuscript to be accepted for publication after minor corrections are done.

Author Response

Dear Reviewer 2,

Thank you for your time and positive attitude in evaluating our study. We have found the comments of the Reviewer 2 very useful, and we hope that we can manage to reply to them satisfactorily and the changes we have made will meet with the approval of the reviewers.

The point-by-pont responses are in attached file.

Kind regards,

Andrea Suranyi
